# Genome-Wide Identification, Characterization and Expression Patterns of the Pectin Methylesterase Inhibitor Genes in *Sorghum bicolor*

**DOI:** 10.3390/genes10100755

**Published:** 2019-09-26

**Authors:** Angyan Ren, Rana Imtiaz Ahmed, Huanyu Chen, Linhe Han, Jinhao Sun, Anming Ding, Yongfeng Guo, Yingzhen Kong

**Affiliations:** 1Key Laboratory for Tobacco Gene Resources, Tobacco Research Institute, Chinese Academy of Agricultural Sciences, Qingdao 266101, China; 2College of Agronomy, Qingdao Agricultural University, Qingdao 266108, China; 3Center for Agricultural Resources Research, Institute of Genetics and Developmental Biology, Chinese Academy of Science, Shijiazhuang 050021, China

**Keywords:** PMEI, gene structure, *cis*-acting element, gene expression, qRT-PCR, cell wall, *Sorghum bicolor*, expression patterns

## Abstract

Cell walls are basically complex with dynamic structures that are being involved in several growth and developmental processes, as well as responses to environmental stresses and the defense mechanism. Pectin is secreted into the cell wall in a highly methylesterified form. It is able to perform function after the de-methylesterification by pectin methylesterase (PME). Whereas, the pectin methylesterase inhibitor (PMEI) plays a key role in plant cell wall modification through inhibiting the PME activity. It provides pectin with different levels of degree of methylesterification to affect the cell wall structures and properties. The PME activity was analyzed in six tissues of *Sorghum bicolor*, and found a high level in the leaf and leaf sheath. PMEI families have been identified in many plant species. Here, a total of 55 pectin methylesterase inhibitor genes (*PMEIs*) were identified from *S. bicolor* whole genome, a more detailed annotation of this crop plant as compared to the previous study. Chromosomal localization, gene structures and sequence characterization of the *PMEI* family were analyzed. Moreover, *cis*-acting elements analysis revealed that each *PMEI* gene was regulated by both internal and environmental factors. The expression patterns of each *PMEI* gene were also clustered according to expression pattern analyzed in 47 tissues under different developmental stages. Furthermore, some *SbPMEIs* were induced when treated with hormonal and abiotic stress. Taken together, these results laid a strong foundation for further study of the functions of *SbPMEIs* and pectin modification during plant growth and stress responses of cereal.

## 1. Introduction

Plant cell walls are composed of high-molecular-weight polysaccharides including cellulose, hemicellulose, pectin and several glycosylated proteins [1]. These complexes with dynamic structures play important roles in several physiological and developmental processes [2]. Pectin is the most complicated polysaccharide component in plant cell walls and mainly contains homogalacturonan (HG), rhamnogalacturonan I (RG I), rhamnogalacturonan II (RG II) and xylogalacturonan (XGA) [2,3]. The ratio between these four polysaccharides is variable in pectin polymers, but HG is the most abundant one constituting about 65% [3]. The HG domain comprises α-D-(1-4)-galacturonic backbone and can be methyl-esterified in general. The pectin matrix is synthesized in a highly methylesterified form in the Golgi apparatus and then secreted into cell walls, where it is de-methylesterified by pectin methylesterase (PME) [3]. Such events may bring changes in the structural and functional properties of cell walls, which is vital to pollen development, pollen tube growth, organ aging, fruit ripening and plant defense i.e., senescence, wounding, biotic and abiotic stress [4,5,6,7,8,9,10]. Moreover, de-methylesterified pectin produces the pectic acid that can bind to extracellular Ca^2+^ to synthesize pectinate, which helps in the stiffening of cell walls and allows the cells to grow gradually.

The fact that PME activity is regulated by the pectin methylesterase inhibitor (PMEI) was first discovered in *Actinidia chinensis* [11]. Since then, seventy-one PMEI-related genes have been annotated in *Arabidopsis thaliana*, while some of them have been identified to inhibit the PME activity in vivo [12,13,14]. A total of 152 amino acid (aa) residues of PMEI were purified from kiwi fruit through protein chemical analysis techniques. PMEI is a glycoprotein, proposing a much more acidic nature of amino acids in protein sequences. It is active against plant PMEs to form a 1:1 non-covalent complex, while the stability of the complex is strongly pH-dependent [15,16]. In *Arabidopsis*, *AtPME17* is regulated by *AtPMEI4* and *SBT3.5* (subtilisin-like serine protease), which is being involved in mediating the changes in pectin properties and accurate root growth [17]. Furthermore, detailed biochemical characterization verified that *AtPMEI7* regulates the activity of *AtPME3* in a pH-dependent manner [15]. The interconversion between PME and PMEI usually determines cell-cell adhesion, cell wall porosity and elasticity. It also acts as a source of signaling molecules, released upon cell wall stress. Therefore, the identification of PME and PMEI family members in different species would make great biological significances and benefits for fully understanding the mechanism of cell wall modification.

Recent studies demonstrated the crucial role of PMEI in the proclamation of cell and organ size, cell growth acceleration [18] and pollen tube development [5,6,8]. The overexpression of *AtPMEI3* in transgenic lines in *Arabidopsis* exhibited higher methyl esterification of HG in the meristem tissues and exaggerated the phyllotaxis pattern of leaves [19]. Additionally, the overexpression of *AtPMEI5* in *Arabidopsis thaliana* caused twisted stems, organ fusion and increased germination rate [20]. Whereas, AtPMEI6 has been identified as a necessary protein involved in the regulation of mucilage synthesis and extrusion from seed coat in *Arabidopsis* [14]. In flax (*Linum usitatissimum*), 83 gene members of the PMEI family have been identified, which may participate in the formation and modification of pectin in cell walls to influence phloem fibers elongation [21]. It has been demonstrated that *SolyPMEI* in tomato specifically expresses in fruit development where its protein interacts with fruit-specific SolyPME1 to regulate the degree of methyl esterification [22]. Intriguingly, preliminary biochemical and molecular characterization of *VvPMEI1* in grapevine also elaborated its role in grape berry development [23]. The PMEI purified from kiwi fruit has been utilized to detect residual PME activity, to inactivate PME in fruit products and is also used as an adjuvant in the manufacturing of vegetable-derived products [11,24].

Based on previous researches, *PMEI* genes are not only essential in plant development but also have physiological significance in plant abiotic stress tolerance and disease resistance [25,26]. For example, CaPMEI1 from pepper, which is confined in the xylem tissues of vascular bundles, acts as an antifungal protein [4]. It is well demonstrated that the purified recombinant CaPMEI1 protein not only exhibits an inhibitory effect on PME but also plays a role in providing resistance to bacterial pathogens and tolerance to drought and oxidative stress [4]. The overexpression of PMEI from *Actinidia chinensis* in wheat showed a reduced PME activity and a significant increase in the degree of methylesterification [27]. The transgenic line of *AcPMEI* in wheat also showed improved fungal pathogen resistance by altering the methylesterification of pectin cell wall, which acts as a broad spectrum resistance to diseases [27,28]. Thus, the previous studies demonstrate the significance of PMEI in biological and physiological processes, as well as in stress resistance.

Although PMEI families have been reported in many plant species, but only a few studies have been performed on its role in monocots. Up until now, 37 PMEIs were identified in sorghum in previous studies [29]. To gain better insights into the functional of PMEI family, a systematic comparative analysis at the whole-genome level was performed. In this study, 55 PMEIs protein members in *Sorghum bicolor* were identified and characterized, with addition of 18 new members. The evolution relationships, along with chromosomal and subcellular localization were analyzed. We also detected the gene ontology (GO) category annotations, and protein–protein interaction. The expression patterns of different tissues at different growth and developmental stages were obtained through transcriptome microarray data. The responses of *SbPMEIs* to hormone and abiotic stress were detected by quantitative real-time PCR (qRT-PCR). This study will be supportive for further functional characterization of this gene family in sorghum as well as other monocot crops.

## 2. Materials and Methods 

### 2.1. Plant Materials and PME Activity Analysis

Sorghum (*Sorghum bicolor* (L.) Moench) seeds were germinated at 24 + 1 °C under a 16 hour (h) light/8 h dark cycle and seedlings were then transferred to pots in glasshouse conditions. One week after anthesis, the tissues used in the analysis were collected to extract protein for PME activity analysis.

Total protein extracts were obtained using a One-Step Plant Active Protein Extraction Kit (Sangon Biotech, C510004, Taiwan, China) and calibrated with Easy Protein Quantitative Kit (Bradford; TransGen, DQ101, Beijing, China) using bovine serum albumin (BSA) standard solution and Commas Brilliant Blue solution. The same amount of protein from different tissues were used to analyze. The detailed method was followed as previously reported [12]. Protein extracts (10 μg) in a same volume (20 μL) were loaded in to the 4 mm-diameter wells in 1% agarose gels containing citrus fruit pectin (0.1% (*w*/*v*; ≥85% esterified, Sigma-Aldrich, P9561, Germany), citric acid (12.5 mM) and 50 mM Na_2_HPO_4_, having pH 6.5. The gels were kept overnight at 28 °C followed by staining for 1 h with 0.05% (*w*/*v*) ruthenium red and washed for 4 h in water. The stained gels were photographed and the intensity of staining quantified with ImageJ software [14]. Measurements were performed in triplicate and data were normalized with the tissue stem area set to one.

### 2.2. Identification of SbPMEI Members and Sequence Characterization 

Previously reported AtPMEI in *Arabidopsis thaliana* and OsPMEI in *Oryza sativa* protein sequences were retrieved from The Arabidopsis Information Resource (TAIR: https://www.arabidopsis.org/) and Rice Genome Annotation Project (http://rice.plantbiology.msu.edu/index.shtml), respectively [29]. Using the *Arabidopsis* and rice, full-length PMEI protein and homeodomain amino acid sequences, PLASTP was performed in the sorghum genome database (JGI: https://phytozome.jgi.doe.gov/pz/portal.html). Furthermore, produced Stockholm files by aligning AtPMEI and OsPMEI protein sequences with MAFFT v7 for building a hidden Markov model (HMM) profile, were applied to search the sorghum database [30]. To confirm the presence of the PMEI domain, all the candidate sequences were submitted to Pfam 32.0 (http://pfam.xfam.org/) based on an HMM annotated PMEI domain as PF04043 and were analyzed through SMART (http://smart.embl-heidelberg.de/).

Genome sequences, coding sequences (CDS) and chromosome location of PMEIs were also obtained from the JGI database. Physicochemical properties of each *SbPMEI* gene were calculated by ExPASy ProtParam tool (https://web.expasy.org/protparam/), including the amino acids length, molecular weight (MW), isoelectric point (pI) and open reading frame (ORF). The subcellular locations were predicted using WOLF POSRT II (https://www.genscript.com/psort.html) and Plant-mPLoc (http://www.csbio.sjtu.edu.cn/bioinf/plant-multi/).

### 2.3. Phylogenetic Analysis, Chromosome Location and Tandem Gene Duplication

Multiple sequences alignment of protein sequences of AtPMEIs, OsPMEI and putative SbPMEIs members was performed using Clustal W (https://www.genome.jp/tools-bin/clustalw) under the default setting, while the phylogenetic tree was constructed with MEGA v6.0 using the maximum likelihood (ML) and the neighbor-joining (NJ) method with 1000 bootstrap replicates [31]. The chromosome location information was derived from the JGI database. The online software MapGene2 (http://mg2c.iask.in/mg2c_v2.0/) was adopted to determine the chromosomal location of *SbPMEI* genes. The gene names were nominated according to the position on each chromosome from the top to bottom [31]. Gene cluster was defined as adjacent genes from one gene family separated by two or more genes in a 100 kb region within an individual chromosome and tandem gene duplication is the genes similarity within 70% in one gene cluster [32]. The gene similarities in a gene cluster of sorghum were detected according to EMBOSS (https://www.ebi.ac.uk/Tools/psa/).

### 2.4. Gene Structure, GO Analysis and Protein–Protein Interaction Analysis

The gene structures of the *SbPMEI* genes were illustrated with Gene Structure Display Server (GSDS: http://gsds.cbi.pku.edu.cn/) [33], by comparing CDS sequences with their corresponding genomic DNA sequences. Each SbPMEI, AtPMEI and OsPMEI protein was searched in InterproScan 5 (http://www.ebi.ac.uk/interpro/) for GO category annotation [34]. The predicated protein–protein interaction (PPI) networks were generated from the STRING database [35], by submitting SbPMEI proteins independently and jointly.

### 2.5. Cis-Acting Elements Analysis 

According to the chromosome location of *SbPMEI*s, the 1.5 kb upstream region preceding the start codon of sorghum *PMEI* genes were cut out. *Cis*-acting element analysis in this region was detected using online tool PlantCARE (http://bioinformatics.psb.ugent.be/webtools/plantcare/html/) [36]. Then, the figures were drawn by using Adobe Illustrator software according to the collected information.

### 2.6. Expression Patterns of Sorghum PMEI Genes

Expression data of each *SbPMEI* gene in 47 tissue samples at different growth and developmental stages were obtained from PhytoMine (https://phytozome.jgi.doe.gov/phytomine/begin.do) [37]. With a Perl script, the microarray data were compiled into a database, then analyzed with the HemI program using the Euclidean distance and the hierarchical clustering method of complete linkage cluster to display the relative expression level of sorghum *PMEI* genes.

### 2.7. Abiotic Stress Treatments, RNA Isolation and qRT-PCR Analysis

For abiotic stress treatments, 14 day-old seedlings were transferred from the half-strength of solid Murashige and Skoog (MS) medium to 200 ml half-strength of MS solution, which contained each 100 μM of methyl jasmonic (MeJA), 100 μM gibberellin A3 (GA3), 100 μM of abscisic acid (ABA), 25% (*w*/*v*) polyethylene glycol (PEG, MW 6000) and 250 mM NaCl, 100 mM hydrogen peroxide (H_2_O_2_) [4,38]. The plant shoots were collected after treated 0 h, 2 h, 4 h, 8 h and 12 h, then frozen quickly in liquid nitrogen and stored at −80 °C for RNA extraction.

Total RNA was extracted from each sample using TaKaRa MiniBEST Plant RNA Extraction Kit (Code No. 9769). The first cDNA strand was synthesized from 2 μg total RNA using the TransScript One-Step gDNA Removal and cDNA Synthesis SuperMix kit (TransGen, Code No. AT311) and then diluted 1:40 with ddH_2_O used as templates for qRT-PCR. The TB Green^®^ Premix Ex Taq™ II (Tli RNaseH Plus; Takara, Code No. RR820A, Takara, Japan) was used to detect the gene expression. *EIF4A1* gene (Sobic.004G039400.2) was selected as the sorghum housekeeping gene to normalize the various samples [39]. The gene-specific primers for qRT-PCR were designed with the NCBI Primer-BLAST tool. qRT-PCR assay was performed by ABI 7500 Real-Time System. 2^−∆∆CT^ method was used to analyze the gene relative expression level compared with treatment at 0 h [40]. Primers used in this study are listed in the Appendix A.

## 3. Results

### 3.1. PME Activity Analysis of Six Tissues From Sorghum Bicolor

Inhibiting the enzyme activity of pectin methyl-esterase is the main function of the PMEI protein, by forming a 1:1 non-covalent complex [2,3,11]. In order to investigate the sorghum cell walls undergoes PME-mediated de-methylesterification, the PME activity of six sorghum tissues was detected. Total soluble proteins isolated from six tissues and calibrated using the BSA standard solution. The same amount of proteins in the same volume was used for a PME activity assay. Notably, leaf and leaf sheath possessed relatively higher PME activity, while the lowest activity was detected in stem tissues, suggesting that pectin modification occurred in each tissue with different levels (Figure 1).

### 3.2. Identification, Chromosome Location and Tandem Gene Duplicates of PMEI Genes in Sorghum

Each PMEI protein sequence of *Arabidopsis* and rice were taken for BlastP in JGI database to obtain potential PMEI proteins of sorghum. Redundant sequences were removed manually. On the other way, the HMM (PF04043) profile was searched against the related protein database. To further confirm the reliability of these candidate members, each sequence was analyzed by PFAM and SMART tools to ensure the presence of the PMEI homeodomain and without the PME domain. Finally, 55 members, containing the PMEI domain, were identified as PMEIs in the sorghum whole-genome (Figure 2). Compared with the previous study, 18 more sorghum PMEI members were detected in the whole-genome [29].

Gene family expansion in the genome is very important from an evolutionary point of view of the plant. To identify gene clusters and investigating gene tandem duplicates of *SbPMEI*, the gene map of the chromosome is necessary. We employed online software MapGene2 to determine the chromosomal location of *SbPMEI* genes and the chromosomal map was generated by using local sequence databases of the complete genome. The genes names were nominated as *SbPMEI1*–*55* according to the position on each chromosome from the top to bottom (Figure 2). All of the 55 *SbPMEI* genes were mapped to 10 chromosomes of sorghum, as the genes were distributed among all chromosomes unevenly (Figure 2). Chromosomes 1, 3 and 4 were found to carry over seven *SbPMEI* genes (8, 8 and 10, respectively). Only two genes were located on chromosome 9, *SbPMEI50* and *SbPMEI51* (Figure 2).

Tandem duplicates were defined by the distance between two or more genes to be less than 100 kb (gene cluster) and the sequence similarity of these genes kept within 70% [32]. Fifteen gene pairs have been detected belonging to gene clusters on eight chromosomes except for chromosomes 5 and 9, which might make up tandem duplications (Figure 2). After alignment, through the EMBOSS Needle tool, the similarity of four gene pairs exceeded from 70% (*SbPMEI11* and *SbPMEI12*; *SbPMEI39*, *SbPMEI40* and *SbPMEI41*; *SbPMEI47* and *SbPMEI49*; *SbPMEI54* and *SbPMEI55*), which were regarded as tandem duplicates. Detail information on sequence similarity of paralogous pairs is shown in the Appendix A.

### 3.3. Phylogenetic Analysis and Sequence Characterizations

To investigate the evolution of PMEI proteins, the ML tree was generated based on the multiple sequence alignment of 71 *Arabidopsis*, 49 rice and 55 SbPMEI full-length protein sequences. We found PMEI members fell into five clades (Figure 3A,B). Most members in Clade 4 and Clade 5 belonged to rice and sorghum (monocot clade), while Clade 1–3 (dicot clade) contained a large part of *Arabidopsis* PMEIs. In order to investigate the homology between SbPMEIs, 55 protein sequences were aligned and a phylogenetic tree of all the full-length protein sequences was constructed using the MEGA v6.0 program by the neighbor-joining (NJ) method (Figure 3C). Based on phylogeny, SbPMEIs were classified into three subfamilies (subfamily I, II and III) according to the clusters exhibited on the tree (Figure 3C), while each subfamily contained 20, 18 and 17 members, respectively (Table 1). The protein sequences (Appendix A), CDS (Appendix A) and genomic sequences (Appendix A) for the 72 members were downloaded from the JGI database.

The length of PMEI protein sequences were generally two hundred aa, ranging from 158 aa of SbPMEI14 to 568 aa of SbPMEI33 with an average length of 205 aa, (Table 1). The molecular weight of the 55 SbPMEI proteins ranged from 17.11 to 59.99 kDa. The pI value ranged from 4.15 to 10.66 for all the SbPMEIs, which is absolutely a wide range. Moreover, all the SbPMEI proteins contained signal peptides, ranging from 16 to 57 amino acids in length. Each protein contains a signal position in the 50 aa N-terminal (Table 1), and 22 SbPMEIs contain a transmembrane domains (Figure 4), which is required for targeting to cell walls. WoLF PSORT II and Plant-mPLoc were used to predict the subcellular location, shown that SbPMEI proteins were mainly localized in the cell membrane and extracellular matrix, suggesting that most of PMEI proteins may be involved in the modification of cell wall during the cell wall synthesis and modification processes.

### 3.4. Gene Structure Analysis of SbPMEIs

To investigate the gene homology relationship, the NJ tree was generated of SbPMEI proteins and three subfamilies were identified accordingly (Figure 3C and Figure 4A). Each protein contained the PMEI conserved domain and some also contains transmembrane region in N-terminal to help the protein target on cell walls (Figure 4B). Gene structures of the *SbPMEIs* were obtained by comparing the predicted CDS with their corresponding genomic DNA sequences using GSDS (Figure 4C).

As previous reported, most *PMEI* genes usually contain only one exon in many species, while 1.25 and 1.12 exons on average were detected in *Arabidopsis* and rice respectively [29]. Expectedly, such events were also observed in sorghum, only seven out of 55 *SbPMEIs* genes had two exons and the *SbPMEI* genes contained 1.12 exons on average. Interestingly, *SbPMEI8* had a very long intron region as compared other *SbPMEIs* (Figure 4C). Members of *SbPMEIs* containing intron regions had been found in each subfamily. These results showed that the *PMEIs* in sorghum and rice were more conserved in gene structure than *Arabidopsis* [41]. 

### 3.5. GO Analysis and Protein–Protein Interaction Prediction

In order to further analyze the protein function of SbPMEI, category annotation was analyzed using Interpro Scan 5 (Appendix A). Each sorghum, *Arabidopsis* and rice PMEI protein was annotated with GO: 0004857, which was displayed as “enzyme inhibitor activity” and was defined as “Molecular Function”, suggesting that all mature PMEI proteins have a function of preventing or reducing the activity of an enzyme, which points to PME in plant, and it proved the accuracy of identifying the *PMEI* genes in sorghum (Appendix A).

In order to investigate the co-expression genes with PMEI in sorghum, all members detected the protein–protein interaction. We found that only SbPMEI28 and SbPMEI35 showed an interaction with each other among 55 SbPMEIs. When proteins were checked individually for their interaction networks, SbPMEI18 showed interaction with various proteins with Synaptobrevin and KISc domain, which play important roles in intercellular transport of organelles especially from endoplasmic reticulum to Golgi apparatus (Figure 5A, Appendix A). SbPMEI21 interacted with three PME proteins, which might be the targets of SbPMEI21 during cell wall modification (Figure 5B, Appendix A).

### 3.6. Cis-Acting Elements Detection in Promoter Regions of SbPMEI Genes

In order to further investigate the regulatory mechanism of *SbPMEI* genes, by PlantCARE, up to 88 kinds of *cis*-acting elements were detected in the 1.5 kb upstream regions of the identified *SbPMEI* genes, which might be useful to understand the regulatory cascade of the plant at different development stages under varying environmental factors. They are identified as: Light-responsive, hormone-responsive, environment stress-related, development-related, transcript factors binding sites and some elements of unknown function (Appendix A). Most of those *cis*-elements appeared two or more times in the 1.5 kb promoter region of sorghum *PMEI* genes that was supposed to enhance their binding effects to their corresponding trans-acting factors (Figure 6).

A total of 24 light-responsive elements were detected, being the most abundant type in the upstream regions of *PMEI* genes. Among these, G-box and Box4 were the most abundant than the rest of others, which were detected in the promoters of 41 and 33 *PMEI* genes, respectively, occupying over 60 percent of the *PMEI* family members in sorghum. Moreover, each PMEI member contained one or more kinds of light-responsive elements. Among them, *SbPMEI20* and *SbPMEI45* contained eight kinds of such elements in their promoter regions. Whereas, *SbPMEI5* and *SbPMEI62* only contained G-box and sp1, respectively.

Eleven plant hormone-responsive elements were detected in the upstream of 55 *Sorghum PMEI* genes, including ABRE (involved in the abscisic acid responsiveness), AuxRR-core (auxin responsiveness), GARE-motif, and P-box (gibberellin-responsive element), SARE (salicylic acid responsiveness), TATC-box (gibberellin-responsiveness), TCA-element (salicylic acid responsiveness element), TGA-element (auxin-responsive element), ERE (ethylene-responsive element) and CGTCA-motif/TGACG-motif (involved in the MeJA-responsiveness; Appendix A). CGTCA-motif and TGACG-motif (the most abundant hormone-related elements) appeared at the same site and the opposite chains in the upstream regions of 42 gene members, followed by the ABRE element, which appeared in 40 *SbPMEI* genes. While, SARE appeared only in the promoter of *SbPMEI38*. *SbPMEI12* while *SbPMEI18* only had one copy of TGA and TCA-element, respectively. Whilst, *SbPMEI24* had two repeats of GARE-motif. Eight types of hormone-related elements were found in the promoter region of *SbPMEI45,* containing the most kinds of hormone elements.

The environmental stress-responsive elements were another important *cis*-acting type elements, which were detected in six different kinds. Unlike light and hormone-responsive elements, six *SbPMEIs* did not contain such elements in the promoters. As the most abundant environment-related element, ARE appeared in 39 members of *PMEI* family, especially in subfamily I and subfamily III, which is related to the anaerobic induction. Followed by MBS, a MYB binding site involved in drought-inducibility, occupied 27 *SbPMEI* genes.

Furthermore, the development-related element i.e., CAT-box was found to be located in the upstream regions of 19 *SbPMEIs* promoters, suggesting that these genes might involve in cell wall methylesterase/de-methylesterase modification during cell division. NON-box was another detected *cis*-acting regulatory element related to meristem specific activation, which only appeared in the promoter region of *SbPMEI24*. Sixteen *PMEI* genes contained the seed-specific regulation element in their promoter regions, RY-motif, suggesting their role in seed development.

### 3.7. Expression Patterns of PMEI Genes in Growth Lifecycle

The RNA-seq online data was used to analyze the *SbPMEI* genes expression patterns in 47 different tissues at different developmental stages throughout the sorghum lifecycle [37]. Heat map depicted the hierarchical clustering that was further used to analyze the expression profiles of 55 *SbPMEI* genes (Figure 7).

The expression patterns of *SbPMEIs* were divided into six distinct clusters (A–F), according to the heat map (Figure 7). Cluster A comprised of two genes, *SbPMEI54* and SbPMEI55, belonging to subfamily II, showed highly expressed in internode than other genes and low expression in leaf. Whereas, cluster B consisted of thirty-four genes, which showed a very high expression level in panicle. While, *SbPMEI11*, *SbPMEI12*, *SbPMEI14*, *SbPMEI27*, *SbPMEI45* and *SbPMEI46* were found as the panicle-special expression genes. Genes in cluster C were highly seed related expressed genes, including *SbPMEI15*, *SbPMEI24*, *SbPMEI25*, *SbPMEI31* and *SbPMEI48*. Four genes constituted the cluster D and E, mainly imploded in leaf and shoot tissues. While cluster F, showed high accumulated level in stem and roots.

### 3.8. Expression Patters of SbPMEIs under Exogenous Hormones and Abiotic Stresses 

Based on the analysis above, eight *SbPMEIs* were selected randomly to analyze the expression pattern under exogenous hormones or abiotic stress treatment, including MeJA, GA3, ABA, H_2_O_2_, PEG 6000 and NaCl, using qRT-PCR, compared with treated at 0 h (Figure 8). 

For exogenous hormone treatments, most *SbPMEIs* were found to be up-regulated, especially, under MeJA and GA3 treatments. The expression of *SbPMEI2*, *SbPMEI19* and *SbPMEI20* were significantly induced by MeJA and reached a peak after treated at 8 h and 12 h, respectively (Figure 8A–C). For GA3 treatment, most of the SbPMEI transcript accumulated after 2 h, reaching a peak after 4 h and declined thereafter. The expression level of *SbPMEI4* was significantly reduced when treated by GA3. ABA strongly increased the expression of *SbPMEI2* at 2 h and then declined gradually. While *SbPMEI20* was observed at a higher level at 2 h that was slightly reduced after 4 h, then peaked at 8 h. All the expression compared with control.

Abiotic stresses analysis showed that NaCl stress induced the expression of detected *SbPMEIs*, especially, *SbPMEI20* and *SbPMEI35*. The expression of the two genes were induced at 2 h and got a peak at 4 h, then reduced thereafter (Figure 8E–G). For H_2_O_2_ treatment, *SbPMEI35* was the most sensitive gene to H_2_O_2_ amongst the detected *SbPMEIs*. Most of the detected genes were expressed highly at 8 h in response to H_2_O_2_ stress. Furthermore, the expression patterns of *SbPMEIs* that responded to drought stress treated by PEG, were induced at 2 h as compared with control (Figure 8F), suggesting that *SbPMEIs* were sensitive to drought. However, the expression level of *SbPMEI34* was decreased at 12 h, when treated by PEG. Taken together, the *PMEI* genes might play important roles in the stress response.

## 4. Discussion

PMEI proteins play an important role in pectin modification of cell walls. Since the first report in kiwifruit [11], PMEI family members have been identified in many plant species, including *Triticum aestivum* [38], *Brassica oleracea* [31], *Linum usitatissimum* [21] and so on. PMEI was found to be active against the activity of PME, to inhibit the cell wall de-methylesterified processes by forming a 1:1 complex. The identification of PMEI family is necessary to understand the modification of plant cell walls. Previous studies have demonstrated that PME activity affects the cell wall structure and plant phenotype, which is mainly regulated by pH and PMEI [2,3]. The increase of PME activity in the *pmei6* mutant, resulted in the defective seed coat mucilage releasing [14]. Our previous study has reported that transcript factor MYB52 can activate the expression of *PMEI14* to inhibit the PME activity in seed coat mucilage [12]. In this study, we monitored the PME activity in six sorghum tissues at the stage of one week after anthesis, which showed a great difference in different of tissues, e.g., lowest activity was detected in stem (Figure 1), that is quite contrary with the previous report of the highest level in fruit followed by stem tissues in tomato [42]. This indicates that PME activity varies between species. However, the high relative PME activity in leaf and leaf sheath might partly result from the low expression level of *SbPMEIs* in these tissues in Cluster A–D and F (Figure 1 and Figure 7). Although most of the *SbPMEIs* were highly expressed in panicle at the stage of anthesis (Figure 7), the PME activity detected in panicle at the stage of anthesis after one week was still relatively high (Figure 1), suggesting that the PME activity might be differently regulated by PMEI transiently or regulated by complicated factors. The molecular mechanism of PME and PMEI should be further studied to explain the methylesterified modification of cell wall in different tissues of developmental stages in sorghum. When protein–protein interaction were detected by searching each SbPMEI individually, SbPMEI18 showed interaction with various proteins involved in intercellular transportation from endoplasmic reticulum to Golgi apparatus, which indicated that it might perform function in cell walls (Figure 5A). SbPMEI21 showed interaction with three PME proteins in sorghum, suggesting that SbPMEI21 may target one of those three PME to modify the cell wall through generating 1:1 non-covalent complex (Figure 5B, Appendix A).

For decades, PMEs and PMEIs have been identified and reported in many species, 66 PMEs and 71 PMEIs in *Arabidopsis* [29], 79 PMEs and 48 PMEIs in tomato [42], 43 PMEs [43] and 49 PMEIs [29,44] in rice. According to previous report, 37 PMEIs and 23 PMEs were identified in *Sorghum bicolor* [29], and the number of PME family was confirmed [45]. However, in this study, a total of 55 PMEI proteins with PMEI conserved domain were identified in sorghum, using 49 rice and 71 *Arabidopsis* PMEI protein sequences to performing blastP in the JGI database (Figure 3, Table 1). The SMART and Pfam Tool were used to confirm each putative PMEI protein containing the PMEI domain (Figure 4B). Moreover, GO annotation analysis depicted that, each SbPMEI involved in the enzyme inhibiting activity (Appendix A). According to the phylogenetic relationship, PMEIs in sorghum and rice clustered together, rather than with *Arabidopsis* (Figure 3A,B). Furthermore, the same event happened in gene structures (Figure 4C), containing 1.12 exons on average, same with rice and less than *Arabidopsis* with 1.25 exons on average, suggesting the gene structures in the monocot was more conserved. The sorghum PMEI members clustered into three subfamilies according to the NJ phylogenetic tree, which was supported by the gene structures, *cis*-acting elements and the expression pattern results (Figure 3, Figure 4, Figure 6 and Figure 7). All of the genes were distributed among all ten sorghum chromosomes unevenly (Figure 2).

Gene expression pattern provides important clues for gene function, associated with the divergence of the promoter regions [31,46]. *Cis*-acting elements contained in the upstream region of genes play an important role in gene expression. For example, the transcriptional factor MYB52 belonging to the R2R3-MYB family, negatively regulates pectin de-methylesterification in seed coat mucilage, by binding the site of AA(A/C)AAAC in the promoter of *PMEI6* and *PMEI14* in *Arabidopsis* [12]. Therefore, we analyzed the *cis*-acting elements in the 1.5 kb upstream regions of the translation initiation codon of each sorghum *PMEI* gene, using PlantCARE. Six types of such elements were found in the promoter region (Appendix A) and four types of functional elements were depicted and analyzed (Figure 6). Light-responsive elements were found more abundant with one of four most important cis-element types, while a total number of 24. Among these, G-box was found in 41 promoters of *SbPMEI* genes. Besides this, eight kinds of light-responsive elements were also detected in the promoter regions of two *SbPMEI* genes (*SbPMEI20* and *SbPMEI60*). However, little is known about the relationship between the expression of PMEI genes and light. In addition, eleven elements were involved with hormone treatment i.e., ABA, auxin, SA, MeJA, ethylene and GA, which is consistent with previous studies about *PMEI* genes in *Brassica campestris* [46]. The CGTCA/TGACG-motif (involved in MeJA) was found in 42 promoters, where ABRE appeared in 40 promoters, suggesting the large number of *SbPMEIs* might be induced by hormones. That is supported by *Capsicum annuum*, *CaPMEI1* was induced by ABA, H_2_O_2_, PEG as well as cold [4]. Furthermore, most of the hormone-responsive elements appeared in the promoter of *SbPMEI45*. Up to 19 such elements were found in the promoter of *SbPMEI4*, out of which nine were related to ABA, eight with MeJA, while each element was related to GA and ABA (Figure 6). While, *SbPMEI4* could be truly induced by MeJA and suppressed by GA and ABA, when treated by exogenous hormones (Figure 8A–C). Moreover, six different kinds of elements were detected involving the environmental response to anaerobic, anoxic, low temperature, drought, defense and wound. ARE involved in anaerobic, presented in 39 promoters. Additional, the MBS *cis*-element (MYB binding site involved in drought) was found in the promoters of 27 *SbPMEI* genes, where about half of the genes were similar to *Brassica compestris* [46]. *SbPMEI20* was significantly induced by exogenous hormones, NaCl and PEG, containing 4 ABA, 12 MeJA and 2 MBS related elements in the promoter (Figure 6 and Figure 8F), proposing an important role in hormone signaling and abiotic stress defense. Furthermore, as the most abundant element related to development, CAT-box, involved in meristem expression, was found in 19 promoters of *SbPMEIs* while 16 genes contained the RY-motif (seed-specific regulation element) in their promoter regions. qRT-PCR analyses were performed for three exogenous and three abiotic stress treatments, to analyze the expression profiles of eight *SbPMEI*s. However, only *SbPMEI2*, *SbPMEI20*, *SbPMEI43* and *SbPMEI35* were significantly induced by ABA treatment, similar to the *CaPMEI1* [4], indicating that the expression profiles of most *SbPMEIs* differed among various treatments.

Moreover, the expression patterns of 55 *SbPMEI* genes were analyzed in 47 sorghum tissues at different growth and developmental stages with a heat map (Figure 7). The expression patterns were divided into six distinct clusters. Genes from the same cluster showed similar expression levels associated with different tissues or organs as well as different stages. Furthermore, the expression patterns differed among most *PMEI* genes in each subfamily, which might functional differentiation after gene duplication events [30]. Members in cluster A, showed a high expression level in the internode, with a lower level in leaf and leaf sheath tissues at any developmental stage. Thirty-four *SbPMEI* genes of cluster B, showed high expression levels in panicle tissues while most of them showed relatively low expression in other tissues (Figure 7). Moreover, six genes were found as the panicle-special expression genes. That high expression level of over half PMEI family members in panicle did not result in low PME activity as expectedly (Figure 1), suggesting the regulation mechanism of PME activity might be complicated. Contrarily, fewer *PMEI* genes expressed highly in leaf and leaf sheath tissues, which might be the reason of a high-level PME activity in those tissues (Figure 1 and Figure 7). However, no gene was detected expressing in all of the 47 tissues, which suggested that the expression patterns of *PMEI* genes were diverse.

## 5. Conclusions

Conclusively, a total of 55 *PMEI* genes in *S. bicolor* were identified and their molecular characterizations were analyzed. Molecular phylogeny of SbPMEIs along with 71 *Arabidopsis* and 49 rice PMEI members revealed that the gene evolution event occurred between monocot and dicot. Futhermore, the PMEI gene structures of sorghum and rice were more conserved than *Arabidopsis*. Each *SbPMEI* contained many *cis*-acting elements in their promoter region, revealing that the expression was controlled by a variety of factors i.e., light, plant hormone, stress, as well as transcriptional factors. Protein–protein interaction prediction shown SbPMEI21 interacted with three PME proteins, which might be the target of PMEI21 to generate 1:1 complex. The expression patterns of analyzed *PMEIs* using transcriptome data revealed that most of the *SbPMEIs* were highly expressed in panicle. However, though most *SbPMEIs* expressed highly in panicle, the PME activity in panicle was not detected as the lowest one comparing other tissues as expected. While a low expression level was detected in leaf and leaf sheath tissues, resulted in the high PME activity. In addition, the expression profiles to the exogenous hormone and abiotic stress treatment by qRT-PCR revealed that the majority of the *PMEIs* were induced by these stresses. Taken together, all the experimental and computational data analysis in this study, presented potentially advance understanding of the molecular and functional mechanism of *PMEIs* in *S. bicolor*, which will be fundamental in order to further study the cell wall of a considerably stable crop plant.

## Figures and Tables

**Figure 1 genes-10-00755-f001:**
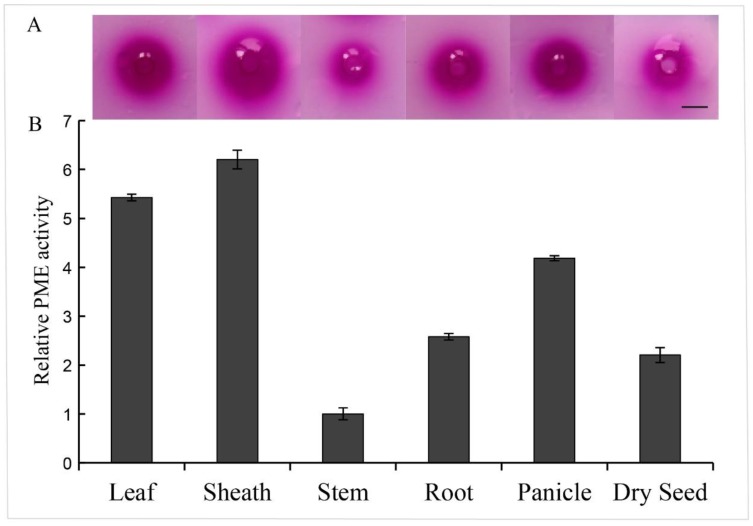
Relative pectin methylesterase (PME) activity analysis of six sorghum tissues. Total protein was extracted from six tissues of sorghum to analyze the PME activity. (**A**) Stained gels of the PME reaction using ruthenium red, the red halo means the de-methylesterificated pectin by PME. Scale bar = 0.5 cm. (**B**) The relative PME activity with the tissue stem area set to one, based on the halo area of stained gel.

**Figure 2 genes-10-00755-f002:**
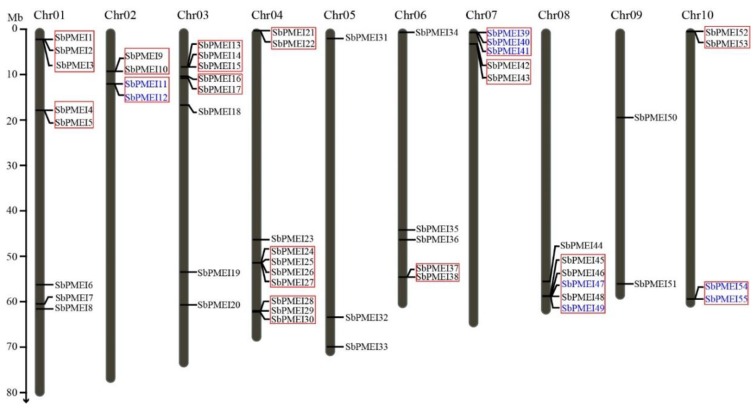
*PMEI* genes’ distribution on sorghum chromosomes. Red rectangles represent the gene clusters, with the distance of *SbPMEIs* in a 100 kb region within an individual chromosome. The tandem gene duplications are indicated by blue colored names. Chr: Chromosome.

**Figure 3 genes-10-00755-f003:**
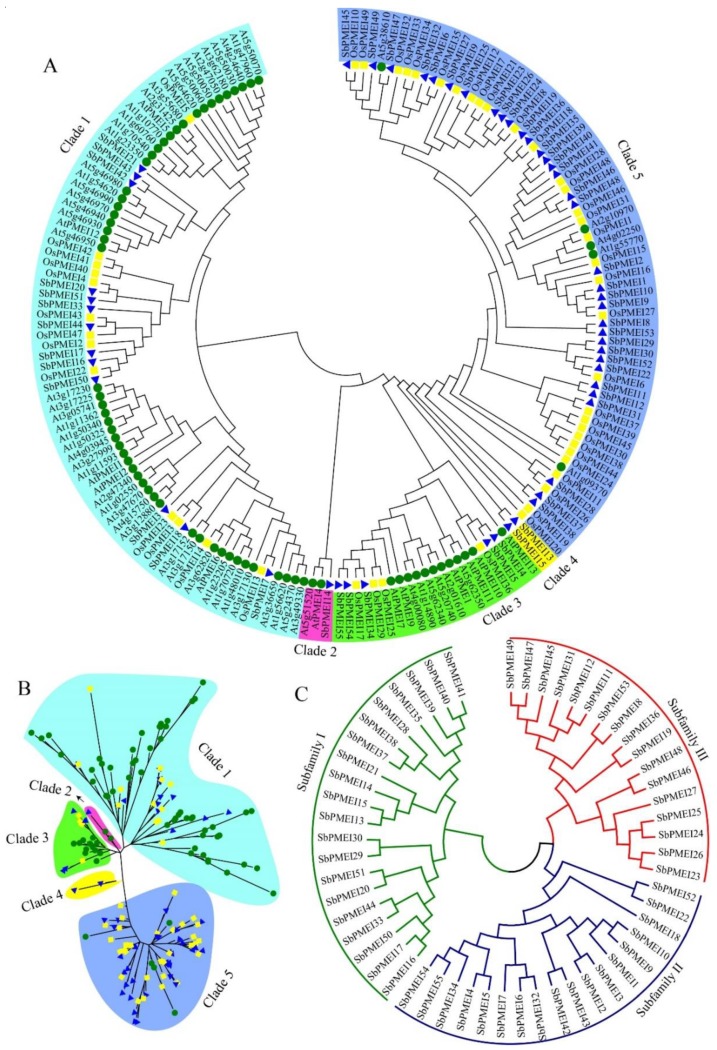
The phylogenetic tree analysis. (**A**) and (**B**) Maximum likelihood (ML) tree of PMEI proteins of *Sorghum bicolor* (blue triangle), *Arabidopsis thaliana* (green circle), and *Oryza sativa* (yellow square) fell into five clades. (**C**) Three subfamilies were evolved based on the neighbor-joining (NJ) tree of 55 *S. bicolor* PMEI proteins.

**Figure 4 genes-10-00755-f004:**
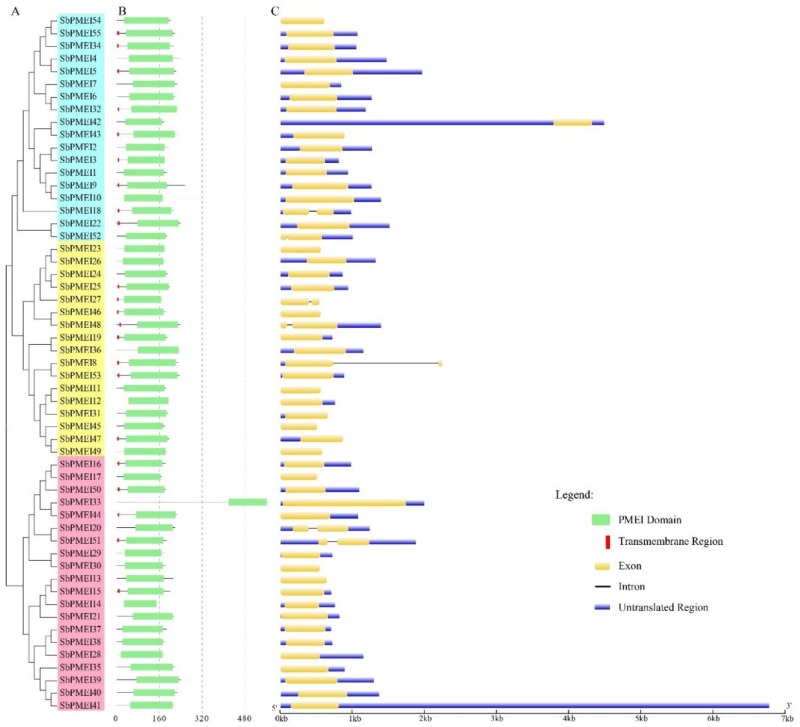
The NJ tree, the conserved domain in 55 SbPMEI proteins and the gene structures of sorghum 55 *PMEI* genes. (**A**) NJ phylogenetic tree by aligning the PMEI protein sequences of sorghum. (**B**) PMEI conserved domains (green rectangle) are presented in each PMEI proteins and transmembrane regions (red rectangle) are presented in 22 PMEI proteins of sorghum. (**C**) Gene structure of each *SbPMEI*. Exons and untranslated regions are presented by yellow and blue boxes, respectively. While, introns are presented by black lines.

**Figure 5 genes-10-00755-f005:**
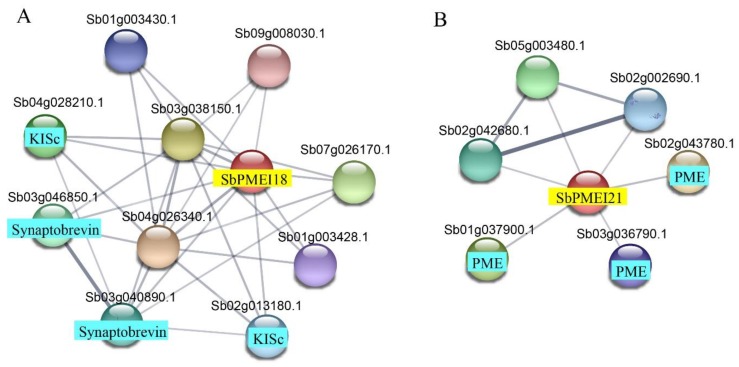
Protein–protein interaction networks of SbPMEI18 and SbPMEI21. (**A**) The interaction network of SbPMEI18. (**B**) The interaction network of SbPMEI21. The networks were generated from the STRING database. The red balls represented the queried protein while the other balls were the identifier. The annotation of the predicted protein was derived from UniProtKB (https://www.uniprot.org/). Line thickness indicates the strength of data support.

**Figure 6 genes-10-00755-f006:**
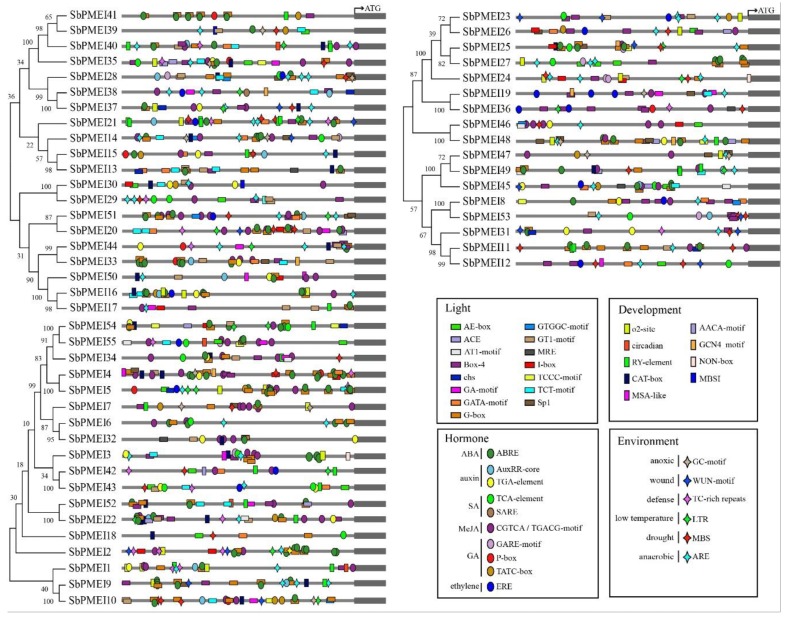
*Cis*-acting elements in the 1.5 kb promoter region of each *PMEI* gene of each subfamily in sorghum. 1.5 kb region before translation start codon (ATG) of each *SbPMEI* gene was analyzed by plant PlantCARE, and four major types of *cis*-acting elements were drawn above accordingly. The NJ phylogenetic tree was made with the proteins from the same subfamily.

**Figure 7 genes-10-00755-f007:**
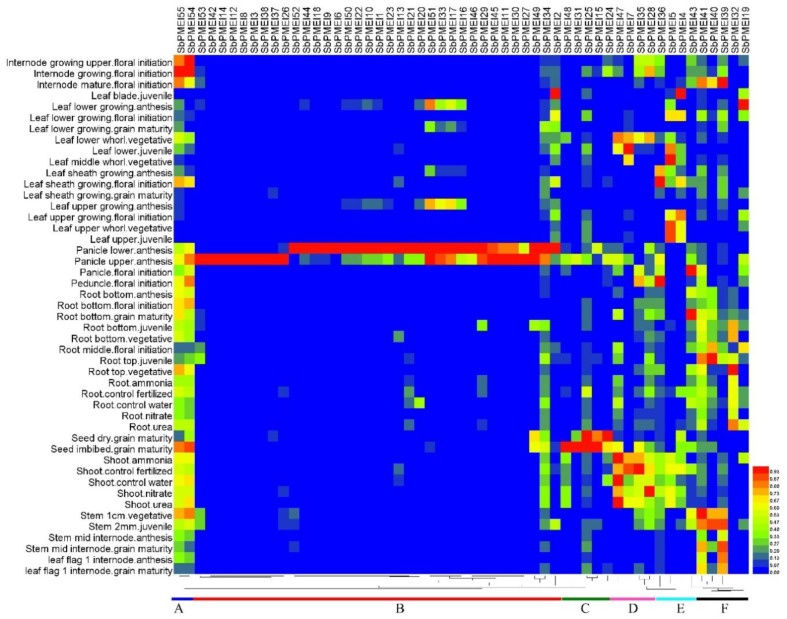
Heat map of 55 *PMEIs* in 47 tissues at different growth and developmental stages. The heat map shows the relative abundances of expressed genes. It ranges from low abundance (blue) to medium abundances (green) and high abundance (red). According to the expression patterns, six clusters (**A**–**F**) were classified.

**Figure 8 genes-10-00755-f008:**
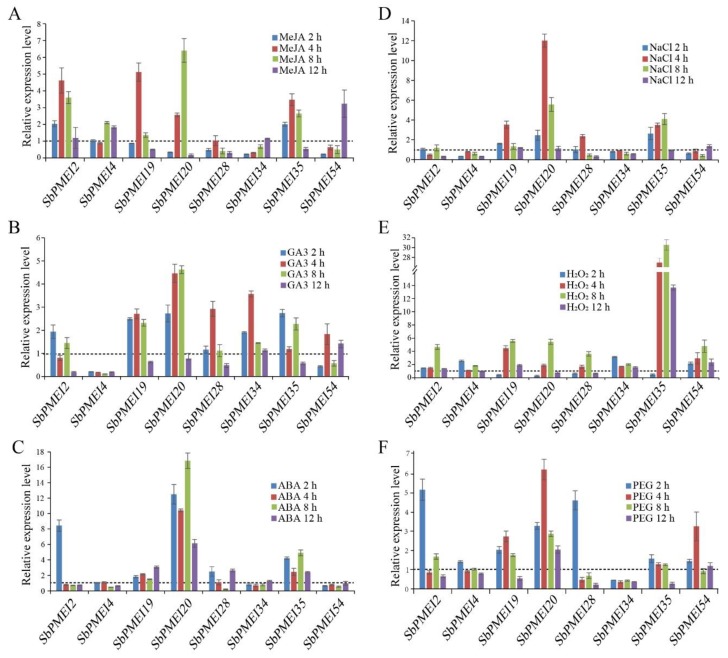
Relative expression levels of eight *SbPMEIs* involved in three exogenous hormones (**A**–**C**) and abiotic stresses (**E**–**F**) using qRT-PCR. Expression patterns treated by: (**A**) 100 μM MeJA; (**B**) 100 μM gibberellin A3; (**C**) 100 μM of abscisic acid; (**D**) 250 mM NaCl; (**E**) 100 mM hydrogen peroxide and (**F**) 25% (*w*/*v*) polyethylene glycol. The relative expression level of eleven *SbPMEIs* were calculated relative to treatment at 0 h.

**Table 1 genes-10-00755-t001:** Characteristics of fifty-five PMEIs identified in the *S. bicolor* genome.

Gene Name	Sequence ID	Subfamily	Signal Position	ORF (bp)	Subcellular Localization		Predicted Features			No. of Intron
					Plant-mPLoc	WOLF PSORT	Length (aa)	NM (kDa)	pI	
*SbPMEI1*	Sobic.001G024300	II	1-24	564	C.M	chlo: 9, extr: 4	187	19.30	5.48	0
*SbPMEI2*	Sobic.001G024400	II	1-31	588	C.M	chlo: 9, extr: 4	195	20.85	4.68	0
*SbPMEI3*	Sobic.001G024900	II	1-34	723	C.M, Nuc	chlo: 14	240	24.76	8.86	0
*SbPMEI4*	Sobic.001G198700	II	1-26	720	C.M, Nuc	chlo: 9, vacu: 3, mito: 1	239	24.29	5.09	0
*SbPMEI5*	Sobic.001G198800	II	1-24	672	C.M, Nuc	chlo: 12, mito: 1	223	22.49	5.87	0
*SbPMEI6*	Sobic.001G288400	II	1-23	660	C.M	chlo: 11, extr: 2	219	21.93	8.83	0
*SbPMEI7*	Sobic.001G323801	II	1-20	681	C.M	chlo: 11, nucl: 1, mito: 1	227	22.88	8.72	0
*SbPMEI8*	Sobic.001G328900	III	1-19	714	Nuc	extr: 12, vacu: 1	231	24.31	8.18	1
*SbPMEI9*	Sobic.002G091200	I	1-25	768	C.M	chlo: 13	255	25.91	4.69	0
*SbPMEI10*	Sobic.002G091300	I	1-22	948	Nuc	chlo: 7, extr: 6	315	30.34	4.15	0
*SbPMEI11*	Sobic.002G102800	III	1-23	555	C.M	extr: 5, vacu: 5, chlo: 1, cyto: 1, plas: 1	184	18.64	6.24	0
*SbPMEI12*	Sobic.002G103000	III	1-38	579	C.M	extr: 5, chlo: 4, mito: 2, E.R._plas: 2, plas: 1.5	198	20.15	4.86	0
*SbPMEI13*	Sobic.003G099500	I	1-32	639	Nuc	chlo: 9, mito: 2, vacu: 2	212	23.01	10.39	0
*SbPMEI14*	Sobic.003G099600	I	1-23	477	C.M	chlo: 9, mito: 2, extr: 2	158	17.11	5.72	0
*SbPMEI15*	Sobic.003G100100	I	1-30	603	C.M, Nuc	chlo: 11, vacu: 2	200	21.41	9.32	0
*SbPMEI16*	Sobic.003G114800	I	1-25	552	C.M	extr: 5, chlo: 3, cyto: 3, vacu: 2	183	19.92	9.10	0
*SbPMEI17*	Sobic.003G114900	I	1-22	507	C.M	extr: 9, vacu: 3, chlo: 1	168	18.06	4.75	0
*SbPMEI18*	Sobic.003G147700	II	1-42	594	C.M	E.R.: 2.5, E.R._plas: 2.5, chlo: 2, mito: 2, extr: 2, vacu: 2, plas: 1.5, nucl: 1	213	22.19	4.57	1
*SbPMEI19*	Sobic.003G203900	III	1-27	570	C.M	chlo: 4, extr: 4, vacu: 4, mito: 1	189	19.61	9.50	0
*SbPMEI20*	Sobic.003G270700	I	1-25	657	C.M	chlo: 9, extr: 4	218	22.85	4.62	1
*SbPMEI21*	Sobic.004G002700	I	1-37	642	C.M	extr: 8, plas: 2, golg: 2, chlo: 1	213	23.48	9.00	0
*SbPMEI22*	Sobic.004G002800	II	1-40	720	C.M	chlo: 3, mito: 2, vacu: 2, E.R.: 2, nucl: 1, cyto: 1, plas: 1, extr: 1	239	25.16	4.79	0
*SbPMEI23*	Sobic.004G147700	III	1-25	558	C.M	vacu: 7, E.R.: 3, plas: 2, extr: 2	185	19.22	5.09	0
*SbPMEI24*	Sobic.004G167700	III	1-27	570	C.M	extr: 9, chlo: 2, vacu: 2	189	19.62	5.07	0
*SbPMEI25*	Sobic.004G167800	III	1-35	600	C.M	chlo: 13	199	20.66	5.89	0
*SbPMEI26*	Sobic.004G167900	III	1-22	543	C.M	extr: 13	180	18.85	5.43	0
*SbPMEI27*	Sobic.004G168000	III	1-25	513	C.M	extr: 10, chlo: 1, mito: 1, vacu: 1	170	18.17	5.90	1
*SbPMEI28*	Sobic.004G277100	I	1-19	543	C.M	extr: 6, chlo: 3, vacu: 3, nucl: 1	180	18.79	4.46	0
*SbPMEI29*	Sobic.004G277600	I	1-27	537	C.M	chlo: 8, extr: 5	178	18.87	4.98	0
*SbPMEI30*	Sobic.004G277700	I	1-26	546	Nuc	extr: 12, chlo: 1	183	18.91	5.45	0
*SbPMEI31*	Sobic.005G022600	III	1-23	588	C.M	extr: 11, chlo: 1, cyto: 1	193	19.47	6.09	0
*SbPMEI32*	Sobic.005G160700	II	1-30	696	C.M	chlo: 13	231	23.52	8.42	0
*SbPMEI33*	Sobic.005G212900	I	1-24	1707	Nuc	vacu: 7, nucl: 2, extr: 2, golg: 2	568	59.99	5.26	0
*SbPMEI34*	Sobic.006G004300	II	1-27	645	C.M	chlo: 7, extr: 3, vacu: 3	214	22.83	10.66	0
*SbPMEI35*	Sobic.006G079100	I	1-21	660	C.M	extr: 9, vacu: 3, chlo: 2	219	22.88	8.24	0
*SbPMEI36*	Sobic.006G094000	III	1-57	714	C.M	chlo: 10, mito: 3	237	25.01	9.47	0
*SbPMEI37*	Sobic.006G194800	I	1-21	567	C.M	chlo: 6, extr: 6, cyto: 1	188	19.84	6.42	0
*SbPMEI38*	Sobic.006G194900	I	1-21	534	C.M	extr: 8, chlo: 4, nucl: 1	177	18.27	6.72	0
*SbPMEI39*	Sobic.007G008000	I	1-25	723	C.M, Nuc	chlo: 13	240	23.74	7.69	0
*SbPMEI40*	Sobic.007G008100	I	1-25	681	C.M, Nuc	chlo: 12, cyto: 1	226	22.94	8.32	0
*SbPMEI41*	Sobic.007G008200	I	1-18	660	C.M	chlo: 12, mito: 1	219	21.84	6.99	0
*SbPMEI42*	Sobic.007G036000	I	1-24	537	C.M	extr: 10, chlo: 1, plas: 1, vacu: 1	178	18.77	4.51	0
*SbPMEI43*	Sobic.007G036200	I	1-24	552	Nuc	extr: 11, chlo: 1, vacu: 1	183	20.05	5.19	0
*SbPMEI44*	Sobic.008G129800	I	1-31	684	C.M	extr: 10, chlo: 1, cyto: 1, mito: 1	227	23.45	9.17	0
*SbPMEI45*	Sobic.008G154500	III	1-22	507	C.M	chlo: 10, cyto: 3	180	17.97	5.28	0
*SbPMEI46*	Sobic.008G154600	III	1-19	555	C.M	extr: 11, vacu: 3	184	18.65	4.59	0
*SbPMEI47*	Sobic.008G154700	III	1-26	591	C.M	extr: 8, chlo: 3, cyto: 2	196	19.03	5.71	0
*SbPMEI48*	Sobic.008G154801	III	1-49	711	Nuc	chlo: 5, cyto: 2, mito: 2, nucl: 1, plas: 1, extr: 1, vacu: 1	237	23.96	6.58	1
*SbPMEI49*	Sobic.008G154900	III	1-23	576	C.M	chlo: 5, extr: 5, cyto: 2, mito: 1	191	18.71	4.90	0
*SbPMEI50*	Sobic.009G091600	I	1-30	555	C.M	extr: 7, chlo: 3, vacu: 3	184	19.99	9.24	0
*SbPMEI51*	Sobic.009G216400	III	1-24	567	C.M, Nuc	extr: 8, vacu: 4, chlo: 1	188	20.66	5.83	1
*SbPMEI52*	Sobic.010G006900	II	1-31	567	C.M	extr: 13	188	19.46	5.36	1
*SbPMEI53*	Sobic.010G007600	I	1-26	708	Nuc	extr: 5, chlo: 4, vacu: 4	235	25.08	4.93	0
*SbPMEI54*	Sobic.010G261000	II	1-21	609	C.M	chlo: 6, extr: 3, cyto: 2, vacu: 2	202	20.55	9.35	0
*SbPMEI55*	Sobic.010G261100	II	1-29	654	C.M	chlo: 14	217	21.62	9.60	0

ORF = open reading frame; MW = molecular weight; pI = isoelectric point; chlo = chloroplast; extr = extracellular matrix; vacu = vacuum; plas = plasmid; E.R. = endoplasmic reticulum; E.R._plas = E.R. plasma; cyto = cytoplasm; mito = mitochondrion; golg = Golgi apparatus; C.M = Cell membrane; Nuc/nucl = Nucleus.

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
