# Peer review of "Genome-Wide Identification, Characterization and Expression Patterns of the Pectin Methylesterase Inhibitor Genes in Sorghum bicolor"

_genes, 2019, doi:10.3390/genes10100755_

Round 1

Reviewer 1 Report

In this manuscript titled “Genome-wide identification, characterization and expression patterns of the pectin methylesterase inhibitor genes in Sorghum bicolor”, PMEI genes in S. bicolorwere identified and characterized using published databases. I think this manuscript lacks biological significance, or authors did not explain this point. For example, chromosomal location was displayed in Fig. 2 and its similarity between PMEI genes and ERF transcription factors were explained, although relationship between expression of PMEI and the ERF transcription factors was not explained. 11PMEI genes were selected and their expression was analyzed in Fig. 8, but why these 11 genes were selected is unknown. Thus this manuscript has no clear theme especially in biological or physiological meaning. Unfortunately, this manuscript is not suitable for publication in Genes, I think.

Other comments.

Fig. 1. Were protein samples electrophoresed? Photos of agarose gels should be needed. Relationship between relative PME activity and staining intensity is not clear. Calibration with standard PME should be demonstrated. What is a biological meaning of chromosomal location? L214. Delete “ Interestingly”. It is natural that the length of PMEI connected to PME is longer. Fig. 3 and Fig.4. Proteins in family VI (e.g. PME 53) does not seem to be longer than those in other families. Fig. 4. Please describe the superfamily names in left. L220. The prediction of PSORT is not completely correct. Probably co-expression analysis or more is required to discuss the physiological roles of PMEIs. L249-L250. Do the PMEs in proteins of super family I belong to the same group? Have PMEs in S. bicolor or Arabidopsis been categorized? L251. What is “distribution”? L261 and L263. What are “twenty PMEIs” and “twenty-four PMEIs” in Arabidopsis? Do these proteins belong to superfamily I? The aim of GO analysis is not clear. L271-. DNA motifs do not completely reflect the expression pattern. Fig. 8. Please describe the superfamily of each gene and discuss the biological meanings more. L385-386. What does it mean? L 439, 458-459. Discussion about the relation between PMEI expression and PME activity seems to be inappropriate. Stem showed the least activity. An activity in panicle was higher than stem, seed, and root. Besides no calibration was performed (Fig. 1), nor was expression of PME discussed. L 475. Funding acquisition is not a requirement for authorship.

Reviewer 2 Report

The manuscript entitled “Genome-Wide Identification, Characterization and  Expression Patterns of the Pectin Methylesterase  Inhibitor Genes in Sorghum bicolor” regards Pectin Methylesterase Inhibitor genes (PMEIs)  genes in Sorghum bicolor. The authors identify 72 Pectin Methylesterase Inhibitor genes that classified into six subfamilies according to their phylogenetic relationship. The chromosomal localization, gene structure and sequence characterizations including cis-Acting elements of these 72  SbPMEIs were analyzed. The expression patterns indicate for some of these genes a strong involvement in development and response to abiotic stress in sorghum.

I believe that the scientific content and the knowledge contribution that this work can provide to the scientific community is relevant. The manuscript analyses a class of very important proteins, exploited by plants for the dynamic regulation of pectin, in Sorghum, plant species of strong agri-food and more recently bioenergetic interest. The results can suggest the role of some inhibitors in the growth and response of plants to abiotic stress.

however I believe that the manuscript needs a careful revision of the English language. In particular, a native speaker should check the correct writing of each sentence because some do not have a correct grammar.

My major comment concerns on the number of SBPMEI identified. The authors identify 72 genes in sorghum, including those that they consider having a PME domain. The authors must align themselves with the literature, removing these genes from all their analyses because they are considered from scientific communities to be PMEs with a domain called pro-region, very similar to PMEIs. The same authors can find confirmation in their results since they find a very low homology in the amino acid sequence compared to the other identified genes, as it is known in different organisms.

Thus, the authors must reassess and present all their data excluding the 19 PMEs that present the pro-region domain. Or they may include analyses on these genes but cannot consider them as PMEIs. Furthermore, there is no clear biochemical evidence that the pro-regions of PME are inhibitors and their role is still under debate. Only after this revision and after an extensive discussion of the revised data it will be possible to conclude on the possible role of PMEI genes in Sorghum, which at present instead encompasses some PMEs.

Other comments:

- In introduction there are some scientific papers that are incorrectly placed. The references 17 and 18 must be moved from the line 64 on page 2 to the line 83 of the same page because they concern the involvement of PMEI in plant defence.

- The following references, concerning the involvement of cereal inhibitors in plant defense, should be added in the introduction session.

Volpi C et al Mol Plant Microbe Interact.2011 Sep; 24 (9): 1012-9

Tundo S, et al Mol Plant Microbe Interact. 2016 Aug; 29 (8): 629-39

-In figure 3 , for easier reading of the results it would be useful to add the name of the 14 PMEIs of Arabidopsis already characterized near the Agi code

- the sentence on page 1 line 44 is wrong. Remove "it is methylesterified and"

- the sentence on page 2 line 54, is not correct since the AtPMEI10, 11 and 12 inhibitors have also been shown to inhibit Arabidopsis PME activity. Review the phrase by re-checking all the inhibitors so far characterized in Arabidopsis.

- the concept of the sentence on page 2 line 59-60 is incomprehensible. Restructure by arguing more clearly

Round 2

Reviewer 1 Report

The manuscript has been considerably revised, however, it has not yet reached the level which can be published. Authors should revise a manuscript more carefully, and they also should discuss the results more thoughtfully.

2_Materials and methods_2.1. When (what growth stage) were tissues collected to extract protein? “leaf and leaf sheath may keep a low degree of methylesterification”. Scientific basis is poor, because PME activity at only one time point has been studied. 1. I mentioned “calibration” for enzyme activity, not for protein amount. Vertical axis should be “relative staining intensity”, because the relationship between staining intensity and enzymatic activity is unknown, Legend of Fig.1. “Pctin” should be “Pectin”. 318. “PMEIs in sorghum are conserved in gene structure”. I don’t think so. Six PMEI genes have one intron at different position. 3. What is difference between Clade and subfamily? Do you think that PMEIs in Sorghum are not compatible with those in Arabidopsis and rice? 5A. Do you mean that PMEIs in sorghum and Arabidopsis participate in different biological processes? Probably many researchers think that PMEIs in Arabidopsis also participate in cell wall modification. I can’t understand the subject of this Figure. L495-496, L. 501-503, L 559-560, L585-586, Discussion is not consistent about PME activity in panicle. Figs. 6 and 8. Authors should discuss the relationship between Cis-elements in Fig. 6 and expression pattern in Fig. 8 (e. g. hormone and stress response). Funding acquisition still remains.

Reviewer 2 Report

The manuscript in this version is scientifically correct. The authors responded adequately to the proposed revisions.

I advise the authors to check precisely every sentence because in some cases there are still inaccuracies. For example on page 5 line 202 the sentence"Inhibiting the enzyme activity of Pectin Methyl-esterification protein is the main function of PMEI protein " is not correct. The function of PMEI is to inhibit enzymes that carry out a pectinsde-methylesterification. 

Same thing for English ... also check the legends of the figures carefully.

Moreover in the method lacks some information.

For example...in addition to PLANTCARE, which software the authors used to graph the elements in the FIG.6.

Improve the materials and methods by specifying steps made to obtain the figures presented

Round 3

Reviewer 1 Report

The manuscript has been considerably revised, however, it still needs to be revised. I can’t agree publication of the present version, although I know that authors want to publish it as soon as possible.

  Please write the next version carefully. And please upload 2 text files with or without editing history.

Comments

L48. Delete “ because of HG”. L155. “ofn” should be replaced with “of”, I think. Fig.3. It is a confusing figure.

3-1. In Fig. 3A, blue triangles and yellow squares represent S. bicolor and O. sativa PMEs, respectively, which is inconsistent with the legends.

3-2. S. bicolor PMEs seem to be classified differently in Fig. 3B and C. Clade color should be added in Fig. 3C.

L. 287-288. Literature showing the relationship between the number of exons (or introns) and conservation (or diversity) of gene structure should be cited. L. 436. “ might result from”→”might partly result from” L. 538 ”resulted”→”might result”
